# Shift-invert diagonalization of large many-body localizing spin chains

**Francesca Pietracaprina[1], Nicolas Macé[1], David J. Luitz[2,3] and Fabien Alet[1⋆]**

**1** Laboratoire de Physique Théorique, IRSAMC, Université de Toulouse,
CNRS, UPS, F-31062 Toulouse, France
**2** Department of Physics, T42, Technische Universität München,
James-Franck-Straße 1, D-85748 Garching, Germany
**3** Max-Planck-Institut für Physik komplexer Systeme,
Nöthnitzer Str. 38, D-01187 Dresden, Germany

⋆ alet@irsamc.ups-tlse.fr

## Abstract

We provide a pedagogical review on the calculation of highly excited eigenstates of disordered interacting quantum systems which can undergo a many-body localization (MBL) transition, using shift-invert exact diagonalization. We also provide an example code at https://bitbucket.org/dluitz/sinvert_mbl. Through a detailed analysis of the simulational parameters of the random field Heisenberg spin chain, we provide a practical guide on how to perform efficient computations. We present data for mid-spectrum eigenstates of spin chains of sizes up to $L = 26$. This work is also geared towards readers with interest in efficiency of parallel sparse linear algebra techniques that will find a challenging application in the MBL problem.



# 1 Introduction

The phenomenon of many-body localization (MBL) [1] has recently attracted a lot of attention, as it touches several fundamental issues of the physics of generic closed quantum systems, such as thermalization and transport. We refer to reviews [2, 3] including recent ones [4–6] which highlight the different aspects of the problem: absence of thermalization and transport, slow propagation of quantum information, compact description of localized eigenstates, dynamical consequences for quench experiments, etc. Crucial to the discussion below is the existence of a many-body localized phase where the standard canonical ensemble description and the eigenstate thermalization hypothesis do not work: each eigenstate behaves individually (e.g. is qualitatively differently from nearby eigenstates), and thus individual eigenstates close by in energy have to be resolved in order to study this markedly different behavior in the MBL and thermal phases [2]. It is important to note that a mixed state of several MBL eigenstates would smear out their individual local features.

The richness of this problem is due to the interplay between various ingredients: disorder, many-body interactions, out-of-equilibrium physics or high-energy phenomena. These elements are already difficult to treat individually from a theoretical point of view, and, needless to say, dealing with all of them at once is a formidable challenge. Moreover, due to the increasing number of highly-controlled cold-atom experiments on the topic (see e.g. [7–10]), providing a comparison with and presenting theoretical unbiased predictions is highly desirable.

Numerical simulations thus have been and remain instrumental in understanding quantitatively many aspects of the MBL problem, albeit not in a straightforward fashion. First of all, the problem inherits the complexity of quantum many-body problems with an exponentially growing Hilbert space in terms of system size. It is important to emphasize that the techniques which are standard in the many-body problem to circumvent or mitigate this complexity are usually not efficient for the MBL problem. The need to access individual eigenstates and not mixtures of them as in a canonical ensemble prohibits the naive use of quantum Monte Carlo techniques or high-temperature series expansions, which couple the system to a heat bath from the start. We also emphasize the need to probe "interior" eigenstates at finite energy density, not only at the bottom of the spectrum. Most works focus indeed on the "infinite temperature" limit, which corresponds to eigenstates in the middle of the many-body spectrum. This forbids a direct application of iterative methods (e.g. Lanczos algorithm) which are only efficient for a few extremal eigenstates, whereas deflation techniques become quickly impractical for

more than a few thousand states at the edge of the spectrum. New methods have been proposed that capture eigenstates or dynamics relatively deep in the MBL phase, based on specific properties of eigenstates in this phase, such as low entanglement and the fact that they are eigenstates of quasi-local conserved operators. This includes the extension of matrix-product states methods [11–14], real space renormalization group techniques to excited states [15], as well as unitary flow methods [16, 17] (for a bird's eye review, see Ref. [6]). However, in order to probe the full phase diagram (and not only the MBL phase) as well as to provide reference data for the methods described above, one needs a technique that works in all regimes in an unbiased manner. Another important computational aspect of MBL is the requirement to average over many disorder realizations. Thus one requires a numerical method that can solve "large" problems, in a reasonable amount of CPU time, as it has to be repeated over at least a few hundred realizations.

Full exact diagonalization of the Hamiltonian matrix can access arbitrary eigenstates of a many-body Hamiltonian. It is limited to matrices of size $\sim 50,000$ at most (in principle larger matrices can be reached using parallel diagonalization, in practice the computational effort is too large due to the average over disorder), which in the example presented below corresponds to spin $1/2$ chains of size $L = 18$. In this paper, we advocate the use of *spectral transforms* to transform the interior eigenvalue problem into an extremal eigenvalue problem on which iterative methods based on the Arnoldi or Lanczos algorithm can be used. Several transforms are possible, and we focus on the one that was found to be the most efficient one for the MBL problem, the *shift-invert* (SI) technique, which in its first application to MBL allowed to reach $L = 22$ spins [18]. We provide in Sec. 3.1 a pedagogical introduction to this method, with details on practical implementation in Sec. 3.2 including a code. We then present a detailed benchmark on the scaling of SI on supercomputers in Sec. 4, and finally present results on very large systems (up to $L = 26$ spins) in Sec. 5. Our simulations are performed on the prototypical MBL Hamiltonian, the random field XXZ spin chain, which for sake of completeness we present in the next Sec. 2.

## 2 Description of the problem

We briefly recall the computational properties of the spin chain model which is studied most extensively for numerical simulations of MBL. It was studied early on, e.g. in Ref. [19–21] and many others. This section could be useful for readers interested in the shift-invert technique and its applications but not familiar with this quantum mechanical problem.

The Hamiltonian for the random field XXZ spin chain is given by

$$H = \sum_{j=1}^{L} \left[ (S_j^x S_{j+1}^x + S_j^y S_{j+1}^y + \Delta S_j^z S_{j+1}^z) - h_j S_j^z \right], \tag{1}$$

where $S^\alpha = 1/2\,\sigma_i^\alpha (\alpha = x, y, z)$ with $\sigma_i^\alpha$ Pauli matrices operating on lattice site $i$. We fix the parameter $\Delta = 1$; this value corresponds to the random-field Heisenberg model, which is a generic representative of systems with $\Delta \neq 0$ which have an MBL transition for disorder $h > 0$. Each $h_j$ is a random number drawn uniformly from a box distribution on $[-h, h]$, where $h$ denotes the strength of disorder, a parameter which will vary and strongly affect the physical properties of the eigenstates. This means that for each value of $h$, we have to deal with several instances ($N_d$ 'disorder realizations', $N_d \approx 10^2$ to $10^4$) of H that are to be treated independently. The features of a uniform probability distribution are not crucial here, and in particular for the numerical methods discussed in this paper.

Each of the $L$ spins has two possible projections in the $z$ direction ($\pm 1/2$), giving a total number of $2^L$ configurations. As however this Hamiltonian commutes with the to-

tal spin in the $z$ direction $S^z = \sum_{j=1}^L S_j^z$, it can be block-diagonalized in each $S^z$ sector ($S^z \in [-L/2, -L/2+1, ..., L/2-1, L/2]$). We will mostly consider chains with $L$ even and treat the $S^z = 0$ block of size $\mathcal{N} = \frac{L!}{(L/2)!(L/2)!}$ (and occasionally $L$ odd for which the largest $S^z = 1/2$ block is of size $\mathcal{N} = \frac{L!}{(L/2+1)!(L/2-1)!}$). The actual matrix sizes $\mathcal{N}$ thus scale exponentially with $L$, asymptotically as $2^L/\sqrt{L}$, and are given in Table 1 for the range of L studied in this work.

To implement the Hamiltonian in practice, we use a computational basis given by product states which are eigenstates of all $S_i^z$ operators and labeled by a configuration of their eigenvalues. Each configuration is labeled with a set of $L$ bits (0 and 1), with an equal number of 0 and 1 for $L$ even ("$S^z = 0$" sector) and one extra 1 for $L$ odd ("$S^z = 1/2$" sector). Matrix elements are as follows:

- Off-diagonal elements $H_{cd}$ are equal to $1/2$ iff the configurations $c$ and $d$ differ by a flip of consecutive different bits (that is ...01.... $\leftrightarrow$ ...10....)

- Diagonal elements $H_{cc} = H_{cc}^{(1)} + H_{cc}^{(2)}$ are the sum of two contributions:

  - (1) a repulsion part: $H_{cc}^{(1)} = \Delta/4(N_{...11...} + N_{...00...} - N_{...10...} - N_{...01...})$ where $N_{...bb'...}$ denote the number of occurrences of the pattern $bb'$ in consecutive bits in the configuration $c$,

  - (2) a random field contribution, which will differ from one disorder realization to another: $H_{cc}^{(2)} = \sum_j (h_j \delta_{...0_j...} - h_j/2)$ with $\delta_{...0_j...} = 1$ if there is a 0 bit in the $j-th$ position.

We work here with periodic boundary conditions, meaning that extremal bits are considered as consecutive in the description above.

The Hamiltonian matrix is real, symmetric and sparse: the number of non-zero off-diagonal elements (all equal to $1/2$) is on average $(L+1)/2$ per line, and their total number thus scales as $\mathcal{N}\log(\mathcal{N})$. All diagonal elements are non-zero (except accidentally) and typically scale as $(\Delta/4 + h/2)\sqrt{L}$, which means that at strong-disorder (where the MBL phase is located) the matrix is diagonally dominated. The total number of non-zero elements is also given in Table 1. Published simulations that obtain eigenstates are up to $L = 16$ with full diagonalization and up to $L = 22$ with the shift-invert technique [18].

Table 1: Matrix sizes considered in this work and their number of non-zero elements of H, for chains of size $L$ with periodic boundary conditions. Note that even though the matrix is symmetric, we report here the total number of non-zero elements of the full matrix H.

| Chain length $L$ | Matrix size $\mathcal{N}$ | Number of non-zeros |
|---|---|---|
| 16 | 12 870 | 122 694 |
| 18 | 48 620 | 511 940 |
| 20 | 184 756 | 2 129 556 |
| 22 | 705 432 | 8 834 696 |
| 24 | 2 704 156 | 36 564 892 |
| 25 | 5 200 300 | 72 804 200 |
| 26 | 10 400 600 | 151 016 712 |

The physical phase diagram of this model is as follows: for small values of $h$ (and $\Delta \neq 0$), the eigenstates follow the eigenstate thermalization hypothesis (ETH) [22–26] and have the same local properties as random (Haar measure) vectors in the middle of the spectrum: this

is the ETH or ergodic phase (corresponding to infinite temperature in the middle of the spectrum). At strong disorder $h$, the eigenstates are 'localized' and 'close' to simple product states (in the limit of $h \to \infty$, they are just individual bit states $1001101\cdots$): this is the MBL phase. It is widely believed that for this model these phases are separated by a single phase transition at a critical value $h_c$ [20], and that the value of $h_c$ depends on the position of eigenstates in the spectrum [18]. This position is parametrized by $\epsilon = (E - E_{\min})/(E_{\max} - E_{\min})$, with $E$ the eigenvalue, $E_{\min} = \min[\text{spec(H)}]$ and $E_{\max} = \max[\text{spec(H)}]$ the extremal eigenvalues, and $\epsilon \in [0,1]$. For $\epsilon = 0.5$ (middle of the spectrum) and $\Delta = 1$, the critical disorder was estimated as $h_c \sim 3.7$ based on finite-size analysis with $L = 14$ to $22$ [18].

## 3 The shift-invert technique

### 3.1 Description of the method

The shift-invert method is based on a spectral transformation to access interior eigenpairs of matrices, i.e. eigenpairs with eigenvalues in the center of the spectrum, which are of foremost interest to study thermalization and MBL. Interior eigenvalues of $H$ are exponentially close to each other in terms of system size and therefore very hard to separate. By introducing a spectral transformation, the spectrum can be transformed such that (i) the eigenvalues of interest are situated at the edge of the transformed spectrum and (ii) the spacing to adjacent eigenvalues increases significantly. For the transformed problem, standard methods based on Krylov subspaces (such as the Lanczos or Arnoldi algorithms) can then be employed, and the eigenvalue is trivially transformed back to the original problem. We will only discuss transformations which leave the eigenvectors invariant.

Let us for concreteness assume that we are interested in a few eigenpairs closest to a target $\sigma$ inside the spectrum of the Hamiltonian matrix H: $\sigma \in [E_{\min}, E_{\max}]$. There are two obvious choices for moving the eigenvalues $E_1, E_2, \ldots$ sorted by distance to the target $\sigma$ to the edge of the transformed spectrum:

- *Spectral fold* by the transformation

$$\mathsf{F} = (\mathsf{H} - \sigma)^2. \tag{2}$$

  This transformation moves all eigenvalues of interest to the *lower* edge of the transformed spectrum $\text{spec}(\mathsf{F}) \subset [0, f_{\max}]$ close to 0. Unfortunately, it also *reduces* the distance to adjacent eigenvalues, rendering this transformation in practice not useful, although the transformation is attractive due to its simplicity.

- *Shift and invert* transformation

$$\mathsf{G} = (\mathsf{H} - \sigma)^{-1}. \tag{3}$$

  Here, eigenvalues $E_i$ of H which are slightly smaller than $\sigma$ are transformed to large negative values at the edge of the spectrum of G, whereas eigenvalues slightly larger than $\sigma$ become large positive values at the edge of the spectrum of G. The distance to adjacent eigenvalues is dramatically *increased*, which improves the performance of iterative solvers for eigenpairs from the edges of the spectrum of G significantly. If the target $\sigma$ is accidentally equal to an eigenvalue of H, the matrix becomes singular and the target should be moved slightly.

For the MBL problem, the shift and invert transformation is the only viable method to obtain central eigenpairs of matrices inaccessible to full diagonalization (the spectrum folding

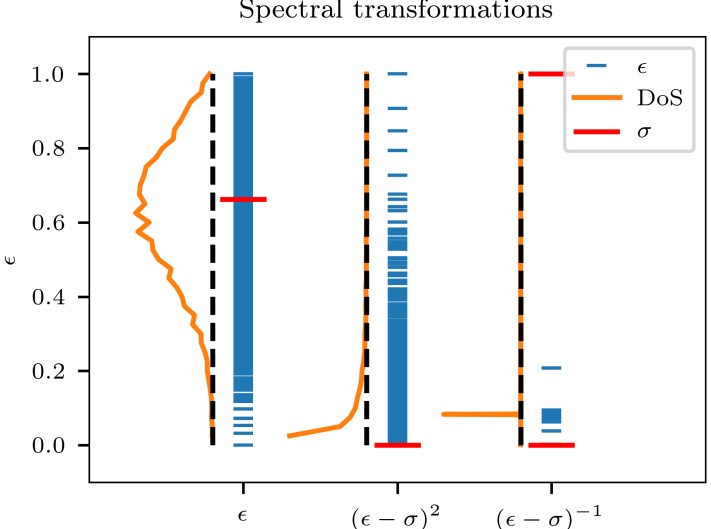

Figure 1: Eigenvalues and densities of states (DoS) for the original spectrum of the clean ($h = 0$) chain of length $L = 14$ in the $S_z = 0$ sector, as well as the transformed spectra, all spectra normalized between 0 and 1.

works, but in our experience does not allow to reach larger systems than full diagonalization). Its major bottleneck is the calculation of the inverse matrix of $(H - \sigma)$. Fortunately, the full inverse matrix is not required, since the Lanczos/Arnoldi algorithms only rely on the repeated product of the transformed matrix $G$ with vectors $\mathbf{x}$. Therefore, the result $\mathbf{y} = G\mathbf{x}$ of the product of $G$ with a vector $\mathbf{x}$ can be expressed as the solution of the linear equation

$$(H - \sigma)\mathbf{y} = \mathbf{x} \tag{4}$$

for the left hand side $\mathbf{x}$. This allows for the elimination of the direct inversion of $H - \sigma$. At this stage, one might try to solve this linear equation using iterative solvers (such as the Jacobi-Davidson algorithm). Unfortunately, the condition number of $(H - \sigma)$ is determined by the ratio of its eigenvalues with maximal and minimal modulus: $\kappa = |E_{\mathcal{N}} - \sigma|/|E_1 - \sigma| \propto \mathcal{O}(L \exp(L))$. Therefore, the problem is severely *ill conditioned* and we can confirm that none of the iterative solvers we tried converges in practice for large enough $L$. In other situations (e.g. Anderson localization [27], see Ref. [28–33] and Sec. 6), iterative solvers can represent the most efficient way to solve Eq. 4 (their effectiveness may depend on the density of states close to $\sigma$).

Therefore, the solution of the linear equation (4) has to be obtained with a deterministic and numerically stable method: the full or partial pivot Gaussian elimination. For this, after reordering with a permutation matrix $P$, an exact LU decomposition in lower and upper triangular matrices $L$ and $U$ of the matrix $(H - \sigma) = LU$ is calculated. As this matrix is sparse, this can be achieved in a massively parallel way with state of the art libraries discussed in Sec. 3.2. With this decomposition, it is then very easy to calculate the solution $\mathbf{y}$ of the linear equation $LU\mathbf{y} = \mathbf{x}$ for any vector $\mathbf{x}$. The main challenge remains the decomposition and storage of triangular factors $L$ and $U$ of very large, distributed sparse Hermitian matrices $H$.

Once the factors $L$ and $U$ are determined, they are used to effectively calculate the matrix vector product $(H - \sigma)^{-1}\mathbf{x}$. This is the central ingredient in iterative eigensolvers, based on the Lanczos/Arnoldi algorithms for obtaining eigenpairs at the edges of the spectrum of very large matrices. The convergence speed of these algorithms is enhanced if the distance to the next eigenvalue is large, which is the case in the transformed spectrum, and hence these methods are efficient. In practice, large numbers of exterior eigenpairs of the order of several hundreds

or thousands can easily be calculated for large problem sizes, using deflation techniques, which project out the already converged outermost eigenpairs $(\lambda, \mathbf{x})$ by subtracting the component $\lambda \mathbf{x}\mathbf{x}^T$ from the matrix.

## 3.2 The implementation in practice

In practice, the best performances for the MBL problem Eq. (1) have been obtained by storing the matrix H in parallel (using the library PETSc [34, 35]), and applying the shift-invert interface of the SLEPc library [36, 37] which allows to use several different iterative eigensolvers (which need the application of G). The choice of using already developed libraries instead of writing a specific dedicated code has several advantages: efficiency of implementation, portability across several platforms and ease of use through the variety of already implemented options and interfaced external packages.

The crux of the method is to efficiently solve the linear system in Eq. (4). After looking at several possibilities and performing different attempts, we found that only two direct parallel solvers were able to reach large sizes for the MBL problem, with excellent performances: MUMPS [38, 39] and STRUMPACK [40, 41]. From the end-user point of view, these solvers have a lot in common: they use a multifrontal procedure to perform the LU factorization, taking advantage of the sparsity of H. The most recent versions of both solvers use hybrid parallelism (shared memory with openMP and distributed memory with MPI), which we can take advantage of. Both solvers also use reordering of matrices using efficient external (parallel) packages, such as METIS [42] (ParMETIS)[1] or SCOTCH (PT-SCOTCH)[2], which can further speed up the calculations and decrease the memory requirement for the largest problems. Finally and quite conveniently, both solvers are interfaced with PETSc.

The main message to be conveyed here is that the bottleneck of the shift-invert strategy with direct solvers is the need to use a very large amount of memory in the factorization step. There are several ways by which we can address the large memory and computational requirements of this problem, namely by using the optimal parameters for each method. We address these questions and provide benchmarks in Sec. 4.

The appendix and the associated bitbucket repository [3] includes a simple C++ code that allows to perform a shift-invert calculation using the PETSc/SLEPc framework as well as option files that interface with MUMPS and STRUMPACK. This code may not be fully optimal which can result in slight time and memory overheads (compared to the benchmarks of Sec. 4), but should be enough to obtain interior eigenstates in the $S^z = 0$ sector of a $L = 20$ chain on a standard workstation with sufficient memory in a reasonable computation time.

## 4 Benchmarks and optimal use of the shift-invert method for the MBL problem

We present in this section benchmarks on time and memory usage of the shift-invert technique described above. The goal here is to present the best strategy for an average practitioner who wants to optimally use given computational resources to produce exact eigenpairs for the MBL problem. We emphasize that the benchmarks are not meant to discuss the scaling of the particular software libraries that we use on our problem, but rather to present the typical time and memory usage that one should expect. The discussion will thus be restricted to sizes $L \leq 22$ (the current state-of-the-art for the method), which can be reached using reasonable

---

[1]cf. http://glaros.dtc.umn.edu/gkhome/views/metis
[2]cf. https://gforge.inria.fr/projects/scotch/
[3]cf. https://bitbucket.org/dluitz/sinvert_mbl

computational resources. Results on even larger $L$ that can be obtained using much larger resources are presented in Sec. 5.

As the system size increases, one needs to use a larger number of parallel resources (using MPI on multiple compute nodes). The larger the number of MPI processes, the faster the factorization and solve phases are performed but, on the other hand, the larger memory *per process* is needed with both MUMPS and STRUMPACK solvers, as a workspace in memory has to be associated to each working process. On parallel supercomputers where compute nodes have several cores sharing the node memory, an optimal strategy can be in some cases to use less MPI processes than available cores on each node: we investigate the use of threads in Sec. 4.1. In order to further reduce memory, we tried approximate factorization (through the block low-rank functionality of MUMPS [43] or the hierarchically semi-separable compression of STRUMPACK [40, 41]) but this did not result in accurate enough results on large systems.

Sec. 4.2 addresses the question of whether the position in the phase diagram influences performances of the shift-invert technique. In Sec. 4.3, we present the breakdown of the execution time in the various steps of the calculation. We finally present in Sec. 4.4 a simple way to reduce memory as well as computational time: the use of *single* precision (instead of the usually chosen *double* precision).

Unless otherwise noted, we perform all computations in double precision and obtain 10 eigenpairs of the Hamiltonian (1) in the middle of the spectrum $\epsilon = 0.5$, where the density of states is close to maximal. For these benchmarks, we used the supercomputer eos (CALMIP, University of Toulouse), whose nodes have 20 processors (2 physical CPUs - Intel Ivybridge E5-2680 v2 at 2.80GHz - with 10 cores each, with a register shared per CPU) and 60GB of available memory, and are interconnected through an Infiniband Full Data Rate network. We use the Krylov-Schur algorithm [44] as implemented in SLEPc to obtain the eigenpairs once the LU factorization is performed.

## 4.1 Optimal strategy and scaling of the computation

We start our analysis by determining the best division of the work between openMP threads and MPI processes: indeed, while the rest of the computation using PETSc/SLEPc does not make use of openMP, both factorization libraries support hybrid parallelization, and, as will be seen in Sec. 4.3 the LU factorization is the most expensive part of the computation (both in time and memory).

A typical MBL calculation involves repeating computations for many disorder realizations (i.e. of the local fields $h_i$), and we find that the optimal strategy is to use the minimal amount of resources per realization of disorder. For chains of $L = 16, 18, 20$, computations fit on a single 60 GB node, while the $L = 22$ system requires a minimum of 6 nodes[4]. In all of our calculations, we use the possibility offered in MUMPS to perform a $LDL^t$ decomposition (which is possible since H is symmetric) rather than a LU decomposition, which impacts favorably speed and halves the memory needed to store the factors.

Table 2 displays our benchmarks results for the typical resources required to perform shift-invert computations, for a varying number of MPI processes and openMP threads. We present results using two different strategies: use, per compute node, either the maximum number of open MP threads, or the maximum number of MPI processes. Let us consider separately the cases of systems of size $L \leq 20$, which fit on a single node, and the case of $L > 20$, which requires the use of multiple nodes and MPI processes. For the first case, we find that the optimal parallelization strategy with respect to CPU time is to use a full shared-memory, openMP

---

[4]For the smaller systems $L \leq 18$, shift-invert computations require less than 3 GB of memory, and thus fit on a single core. The optimal production strategy for these small systems is then to run serially one realization of disorder per core (naive parallelism with no use of MPI or openMP).

Table 2: Typical resources for the shift-invert computation. For the two solvers MUMPS and STRUMPACK, we report real execution time, total CPU time (sum of the execution time over all working CPUs), factorization fill-in ratios (ratio of the number of nonzeros in the factors and in the original matrix to factorize), total memory occupation (as reported by the operating system) and minimum number of 60 GB compute nodes required. For sizes $L \leq 20$ we show results both in the full openMP and in the full MPI setups. For $L = 22$ we use the optimal setup, i.e. a single process with the maximum number of openMP threads for MUMPS (which, for a higher number of processes, requires more memory and thus more compute nodes), and full MPI for STRUMPACK.

| | | MUMPS (maximum openMP strategy) | | | |
|---|---|---|---|---|---|
| $L$ | time (s) | CPU time (s) | fill-in ratio | tot. memory (MB) | nodes |
| 16 | 2.2 | 41 | 98.4 | 204 | 1 |
| 18 | 10.6 | 209 | 264.4 | 947 | 1 |
| 20 | 98.0 | 1957 | 795.4 | 10131 | 1 |
| 22 | 891.6 | 142576 | 2433.8 | 382580 | 8 |

| | | STRUMPACK (maximum openMP strategy) | | | |
|---|---|---|---|---|---|
| $L$ | time (s) | CPU time (s) | fill-in ratio | tot. memory (MB) | nodes |
| 16 | 1.0 | 18 | 88.5 | 264 | 1 |
| 18 | 5.0 | 98 | 226.2 | 926 | 1 |
| 20 | 75.6 | 1509 | 701.3 | 15928 | 1 |

| | | MUMPS (full MPI strategy) | | | |
|---|---|---|---|---|---|
| $L$ | time (s) | CPU time (s) | fill-in ratio | tot. memory (MB) | nodes |
| 16 | 2.0 | 29 | 117.1 | 432 | 1 |
| 18 | 7.1 | 131 | 284.7 | 2807 | 1 |
| 20 | 89.2 | 1772 | 854.5 | 37019 | 1 |

| | | STRUMPACK (full MPI strategy) | | | |
|---|---|---|---|---|---|
| $L$ | time (s) | CPU time (s) | fill-in ratio | tot. memory (MB) | nodes |
| 16 | 1.6 | 20 | 92.6 | 451 | 1 |
| 18 | 6.0 | 108 | 247.5 | 2512 | 1 |
| 20 | 93.8 | 1863 | 701.7 | 21741 | 1 |
| 22 | 607.2 | 72763 | 1978.6 | 243854 | 6 |

setup for STRUMPACK, while for MUMPS a full MPI setup is preferable. With respect to memory, we find that while for a full openMP setup MUMPS has a similar memory occupation as STRUMPACK (actually slightly less due to the use of $LDL^t$), its usage greatly increases with the number of MPI processes; this is less the case for STRUMPACK. The results for the execution time and the total memory usage are shown in Fig. 2 as a function of the number of MPI processes.

As we use more MPI processes (necessary in the case $L > 20$ where multiple nodes and processes are required), we notice that STRUMPACK is both faster and memory efficient with respect to MUMPS for our specific class of matrices (e.g. for a full MPI setup, STRUMPACK has a factor of $\sim 1/2$ for both memory usage and CPU time for $L = 22$ with respect to a MUMPS setup with one MPI process per node). Indeed, with STRUMPACK the computation for $L = 22$ requires around 244GB of memory and fits on 6 nodes, while with MUMPS a minimum of 8 nodes are needed. Note that if the memory occupation was equally divided among the nodes, the computation, e.g. with STRUMPACK, would fit on $\sim 4$ nodes; however we verified that, due

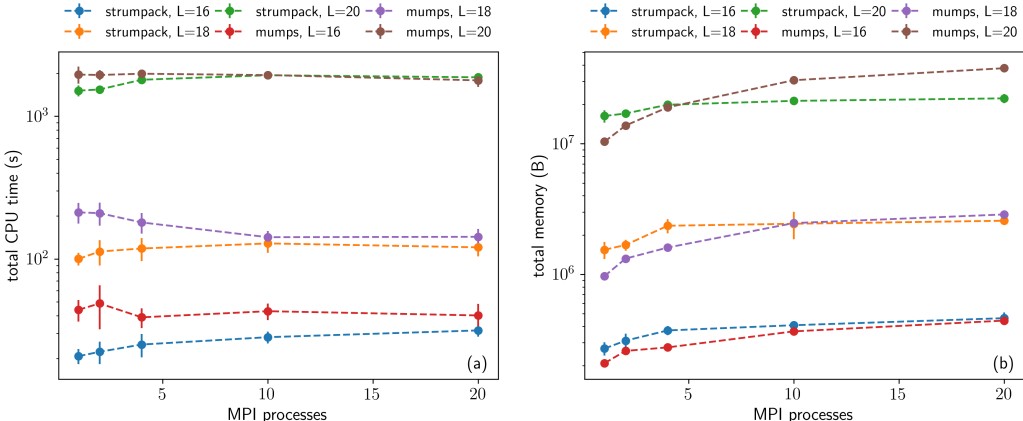

Figure 2: Total CPU time (a) and total memory as reported by the operating system (b) used for shift-invert runs for chains of size $L \leq 20$, which fit on one node. We average all results over a few disorder realizations, although, as we will show in subsection 4.2, fluctuations in the benchmarked quantities are small.

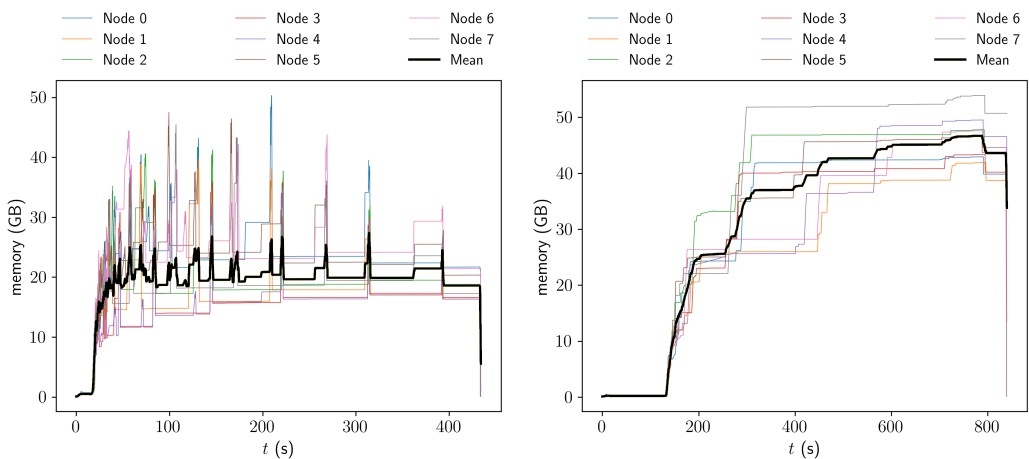

Figure 3: Total memory reported by the operating system of two sample runs using STRUMPACK (a) and MUMPS (b) for a $L = 22$ system. Thin lines correspond to the total memory used on individual nodes, while the thick black line is the average over all nodes.

to load imbalance between the processes, one requires roughly twice the amount of the average memory per node (this also depends on the number of processes/threads and the specific realization). In Fig. 3 we plot the memory usage per compute node as a function of time for two sample runs at $L = 22$ for both STRUMPACK and MUMPS; the imbalance in the memory usage per node is clearly visible. Notice the relatively flat average usage of STRUMPACK with rapid peaks of allocated/de-allocated memory, opposed to the steadily increasing usage of MUMPS.

During the execution of the factorization algorithm, as the matrix elements below the diagonal are eliminated, other elements which originally were zero are filled-in. The *fill-in*, defined as the number of non-zero entries generated by the factorization process, quantifies how sparse the factors LU or LDL$^\mathsf{T}$ are with respect to the original matrix, and is therefore a measure of the difficulty of the factorization. We find that the *fill-in ratio*, that is the fill-in divided by the number of nonzeros in the original matrix, $\mathscr{F} = \mathrm{nnz}_{\mathsf{LU}}/\mathrm{nnz}_{\mathsf{H}}$, is high for

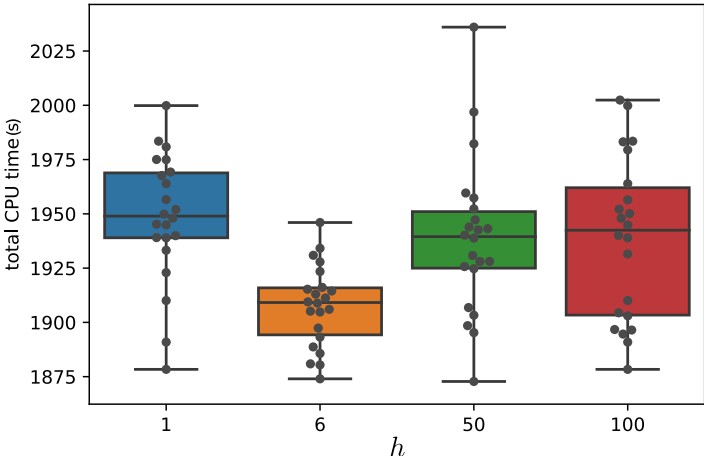

Figure 4: Computation time for various disorder strengths and $L = 20$, using STRUMPACK with a full MPI strategy. For each disorder strength the computation was carried out for 22 disorder realizations, shown as individual points on the plot. The whiskers span the whole time distribution, while the bottom (top) of the boxes correspond to the first (third) quartile. The median value is shown as a solid line.

our class of matrices. It increases rapidly (exponentially) with system size, as can be seen in Table 2 for $L$ up to 22, and in Sec. 5 for larger $L$. Note that since MUMPS explicitly takes advantage of the fact that H is symmetric when using the LDL$^t$ decomposition, the number of nonzero elements that we must consider is halved (as only the upper triangular part of H is used in the algorithm).

## 4.2 Dependence on the disorder strength and on the disorder realization

Since the energy levels in the MBL phase show no level repulsion (opposite to the case of the ETH phase), in principle the problem could be harder depending on the value of the disorder strength $h$. We find that the shift-invert procedure is actually not sensitive to $h$ (at least for a typical number of eigenvalues ranging from 10 to 1000), and is very good at separating the eigenvalues in all cases. In Fig. 4, we show that the execution time is roughly the same for $h$ ranging from small values ($h = 1$ in the ETH phase) to very high disorder ($h = 100$). From Fig. 4 we additionally see that execution times do not fluctuate much between disorder realizations, represented by individual points in the figure. Thus, the shift-invert method is fairly agnostic to the underlying physics that is encoded in the Hamiltonian matrix, making it a powerful and unbiased tool.

## 4.3 Execution time breakdown

We show in Fig. 5 the time spent in the most demanding phases of the computation:

- The partitioning, which corresponds to the reordering of the matrix in order to partition and distribute it with the least amount of off-block elements (for all the tests presented here, we use the METIS package);

- The factorization done by MUMPS or STRUMPACK, which accounts for the largest share of the total execution time;

- The solving step, which corresponds to the iterations to obtain the eigenpairs.

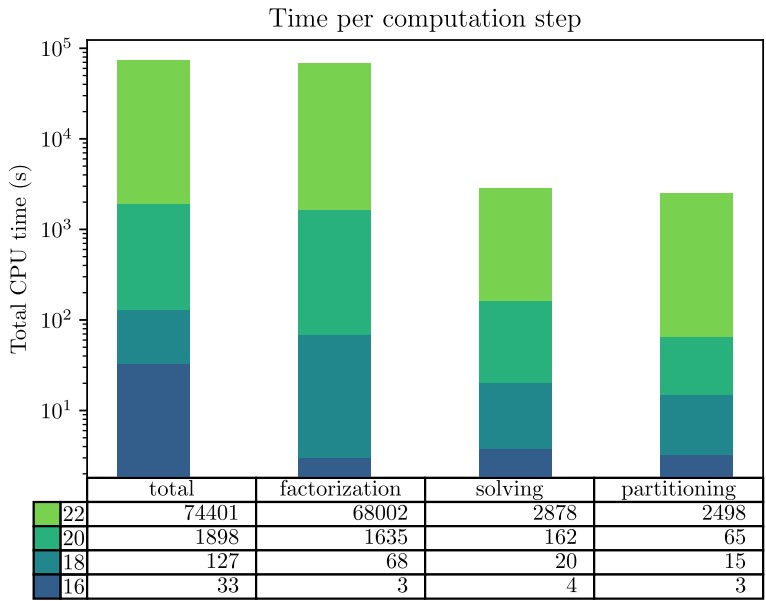

Figure 5: Time per computation step (in *s*), for various system sizes. The execution times are obtained for the computation of 10 eigenpairs using STRUMPACK with the full MPI strategy, and are averaged over 5 runs.

Note that a typical computation also involves (i) constructing and assembling the Hamiltonian, (ii) computing the extremal eigenvalues and possibly (iii) computing observables in the obtained eigenvectors. All these steps occupy only a very small fraction of the total execution time: for instance $< 0.5\%$ for step (i) (around 0.15 s for $L = 20$ and 1.5 s for $L = 22$) and $\lesssim 1\%$ depending on the realization for step (ii). We thus do not include them in the breakdown below as they are essentially negligible.

The use of parallel reordering packages (such as ParMETIS or PT-SCOTCH) can also be useful in the analysis phase of both solvers for very large systems, even though this phase minimally impacts the overall running time in general.

As $L$ is increased, the factorization takes an increasing time compared to the other computation steps. One should however note that the time spent solving depends on the number of eigenpairs one asks for, as shown on Fig. 6. As seen on the figure, for the solving time to be reasonable, one would typically ask for up to 100 eigenpairs.

## 4.4 Reliability of single precision results

A straightforward way to cut the required memory in half is to use single (32 bit) precision instead of double (64 bit) precision for storing real numbers. This can also impact favorably computational time. A number in single precision is precise to at least 6 significant digits (instead of the at least 15 significant digits of a double precision floating point number). For our specific problem, this means that we cannot distinguish anymore two eigenvalues that are within this level of precision. Therefore, when computing physical quantities, these two energy levels can possibly show a spurious hybridization.

In order to understand the possible side effects of performing computations in single precision, we have calculated the local magnetization over an eigenstate $\langle n|S_i^z|n\rangle$ as well as the half-chain entanglement entropy for the same system (with the same realization of disorder) both in single and double precision. The half-chain entanglement entropy $S$, specifically, is chosen as it is very sensitive to the hybridization problems that might arise. It is defined as

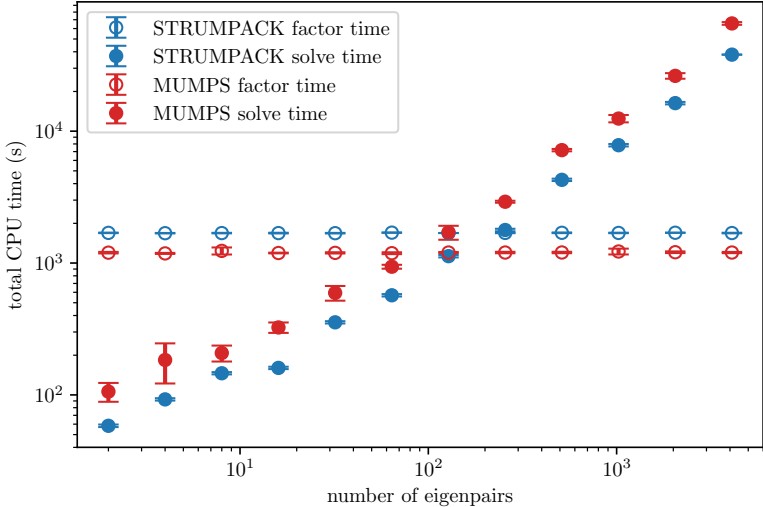

Figure 6: Time required for the computation versus number of eigenpairs requested, for the system of size $L = 20$, using the full MPI strategy. The execution times are averaged over 5 runs.

usual as $S = -\text{Tr}\rho \log(\rho)$ where $\rho$ is the reduced density matrix obtained by tracing out degrees of freedom over half the system.

Fig. 7 compares observables computed in single and double precision, in two cases: (1) at disorder strength $h = 1$, i.e. in the ETH phase, and (2) at $h = 100$, deep in the MBL phase. We can see that in both cases, the local magnetization computed in single precision matches well the magnetization computed in double precision. The entanglement entropy can also be computed satisfactorily well in single precision at low disorder ($h = 1$). At high disorder ($h = 100$) the entanglement entropy computed in single precision matches well the entropy computed in double precision, for most of the eigenstates. In some very rare cases however (last two eigenstates on the right panel of Fig. 7), the entanglement entropy in single precision is overestimated. We argue that this happens when two eigenstates close in energy hybridize, yielding two states of higher entanglement. Numerically, we observe that these hybridization events occur more often as we go deeper in the MBL phase. Note that they remain quite rare for reasonable values of $h$: indeed we do not observe such events for $h \leq 20$ for the system of size $L = 22$, and even very deep in the MBL phase, when $h = 100$, they occur in less than 1% of the cases. One can detect these spurious hybridizations either by running several times the Lanczos/Arnoldi algorithm with different starting vectors (hybridization will occur or not, depending on the choice of the starting vector), or by rotating pairs of eigenstates of neighboring energy in order to minimize their entropy [45]. Fig. 7 shows the corrected entropies obtained using this last method ("rotated single" label). The agreement with double precision data is excellent.

We conclude that one can reliably use single precision to perform the shift-invert method, at least up to size $L \lesssim 24$. By switching to single precision numbers, one halves the required memory, which is extremely beneficial since memory is the limiting factor. We also observed that in single precision the factorization time is approximately halved.

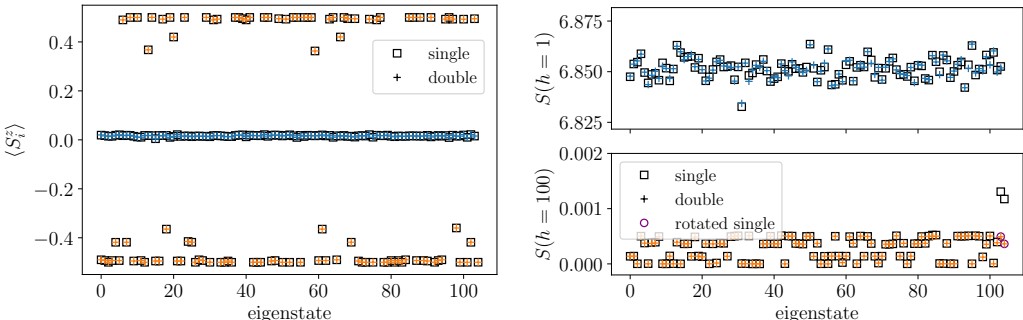

Figure 7: Comparison between quantities computed in single precision (squares) or double precision (crosses), as a function of the eigenstate label (eigenstates ordered by increasing energy), for a system of size $L = 22$, and for $h = 1$ (in blue), and $h = 100$ (in yellow). Left panel: magnetization $\langle S_i^z \rangle$ of the spin $i = L/2$, right panel: entanglement entropy at the $L/2$ cut. The "rotated single" label stands for entropies in single precision corrected by a rotation of the associated pair of eigenstates [45].

## 5 Results for large systems

We now push the methods to its limits by considerably larger sizes, amounting to an important increase in computational resources.

### 5.1 Entanglement entropy and distribution of local observables for $L = 24$

We find interior eigenpairs in double precision for $L = 24$, with typical calculations consisting in computing around 50 eigenpairs at $\epsilon = 0.5$ (middle of the spectrum), using 150 nodes (with 24 cores, model Intel Haswell E5-2680 v3 at 2.50 GHz, 128 GB of memory and a Cray Aries interconnect) of the machine Hazel Hen (HLRS, Stuttgart). The typical fill-in ratio is $\sim 6960$. We use here the solver STRUMPACK, with a full MPI strategy, resulting in typical total execution time of $\sim 14$ minutes for each disorder realization. The factorization by itself takes about 9.5 minutes, with a total performance of 59.5 TFLOPS (as reported by STRUMPACK). We average our results over more than 200 realizations of disorder for each value of disorder $h$.

We first present results for the difference in local magnetization $\delta S_i^z = \langle n|S_i^z|n \rangle - \langle n'|S_i^z|n' \rangle$ between nearby eigenstates $|n\rangle$ and $|n'\rangle$ located at the same energy density $\epsilon = 0.5$. Ref. [46] showed that the distribution of $\delta S_i^z$ was able to capture both the MBL phase and the ETH phase, as well as deviations from the expected gaussian behavior in the later when approaching the transition. We confirm these findings in Fig. 8. At large disorder (MBL phase), a trimodal distribution emerges, which is accounted for by considering that all spins are polarized to their maximum value $\langle n|S_i^z|n \rangle = \pm 1/2$, but independently from one eigenstate to the other. The presence of peaks at $\delta S_i^z = \pm 1$ starts to be substantial at $h \simeq 3.6$. For lower disorder on the other hand, the distribution is peaked around zero, with nevertheless large tails which reveal the non gaussian nature of the distribution [46] (the dotted line in Fig. 8 is the best gaussian fit of the distribution at $h = 2$). These tails are present even for the lowest disorder $h = 2.0$ considered here, and have been attributed to rare regions effects [46].

In Fig.9, we present results on the entanglement entropy for a bipartition ($x = L/4, L - x = 3L/4$) of the chain. It is known that MBL eigenstates exhibit an area law scaling of the entanglement entropy, while extended eigenstates at finite energy density show a volume law [48]. In Ref. [47], a method for determining the scaling of the entanglement entropy as a function of subsystem size was introduced for *single eigenstates* in periodic disordered chains. While for a single bipartition ($[0, \ell), [\ell, L)$), i.e. with a cut before site 0 and

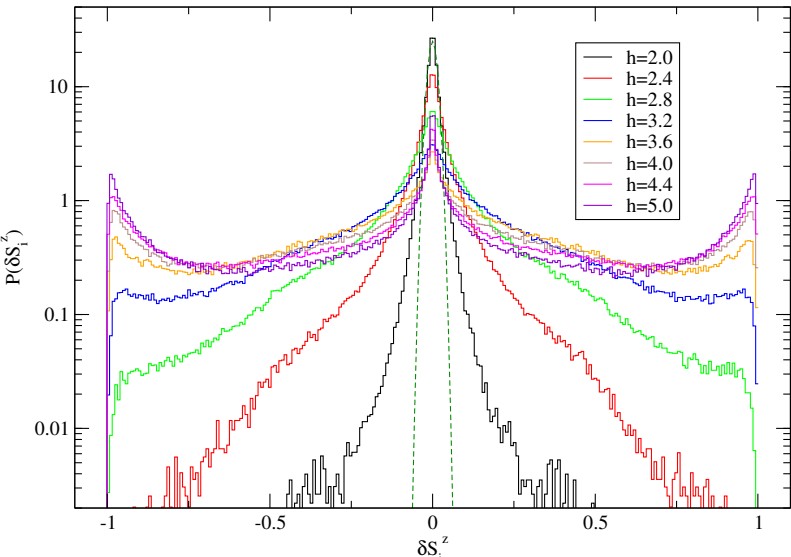

Figure 8: Probability distribution of the difference in local magnetization $\delta S_i^z = \langle n | S_i^z | n \rangle - \langle n' | S_i^z | n' \rangle$ (averaged over all positions $i$) of two nearby eigenstates at the same energy density, for different disorder strengths and chain size $L = 24$. The dotted line is a gaussian fit to the distribution at the lowest disorder $h = 2$.

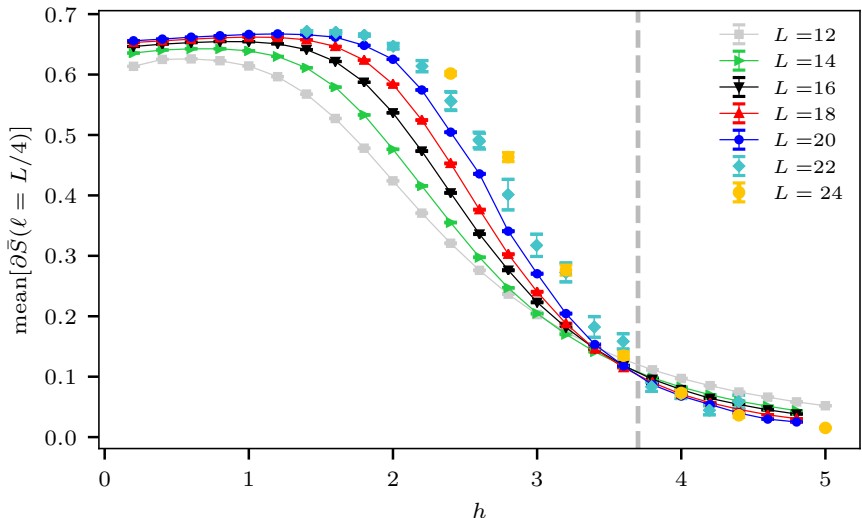

Figure 9: Behavior of the mean of the slope of the entanglement entropy computed for the quarter cut $L/4$ as a function of disorder, for different system sizes. Data for $L \leq 22$ are extracted from Ref. [47].

before site $\ell$ of a chain of length $L$, the entanglement entropy

$$S_{[0,\ell)} = -\mathrm{Tr}\rho_{[0,\ell)} \ln \rho_{[0,\ell)}, \quad \text{with} \quad \rho_{[0,\ell)} = \mathrm{Tr}_{[\ell,L)} |\psi\rangle \langle\psi| \qquad (5)$$

of a wave function $|\psi\rangle$ is not a smooth function of the subsystem size $x$ due to the effect of disorder, it was proven in Ref. [47] that the *average* over all subsystems of length $\ell$ is a smooth and concave function of the subsystem size. This cut averaged entanglement entropy (CAEE) is defined as:

$$\bar{s}(\ell) = \frac{1}{L} \sum_x S_{[x,x+\ell)}. \qquad (6)$$

Note that in this definition periodic boundary conditions are applied, i.e. all positions in the chain are modulo $L$, and the trace in Eq. (5) $\mathrm{Tr}_{[\ell,L)}|\psi\rangle\langle\psi|$ is understood as a partial trace over all degrees of freedom of sites of the chain in the interval $[\ell, L)$.

Following the analysis of Ref. [47] and averaging over all possible cuts (obtained by translation in the periodic chain) for the entanglement entropy in the same eigenstate $|\psi\rangle$, we obtain the cut averaged entanglement entropy $\bar{S}(\ell)$ as a function of subsystem size $\ell$. Since we are interested in how the entanglement entropy scales as a function of subsystem size, we consider the *slope* $\partial\bar{S}(\ell)$ as a function of subsystem size

$$\partial\bar{S}(\ell) = \frac{\partial\bar{S}(\ell)}{\partial\ell}, \tag{7}$$

which is well defined due to the smoothness properties of the CAEE. Since the entanglement entropy as a function of subsystem size exhibits a maximum at the equal bipartition (half chain), we consider a subsystem size close to the quarter cut $\ell = L/4$ where $\partial\bar{S}(\ell)$ is close to maximal, and which is very sensitive to different scaling behaviors (area vs. volume law, cf. Ref. [47]). We estimate the slope $\partial\bar{S}(\ell = L/4)$ of the cut averaged entanglement entropy (using a cubic spline to interpolate for noninteger values of $L/4$) for each eigenstate. As in Ref. [47], we consider the distribution of this slope and represent in Fig. 9 the average of this distribution for a quarter-cut ($\ell = L/4$). The new data for $L = 24$ confirm clearly the well-behavedness of this approach, as different curves (for different sizes $L$) for the average slope appear to cross close to the estimate of the critical point.

## 5.2 Examples of large wave-functions results

We also were able to obtain eigenstates of even larger systems ($L = 25$ and $L = 26$), using single precision. These simulations are very demanding: one disorder sample for $L = 25$ takes $\simeq 14$ minutes on 1000 nodes with a fill-in ratio of $\simeq 12900$ (factorization takes $\simeq 8$ minutes, with a 525 TFLOPS performance), and $L = 26$ requires 2000 nodes of Hazel Hen for an execution time of $\sim 45$ minutes (fill-in ratio $\simeq 23600$, factorization time $\simeq 28$ minutes, factorization performance 1.05 PFLOPS). We thus cannot realistically consider to include these values of $L$ in a finite-size scaling analysis of the MBL transition. However, we expect this could be possible in the near future with higher computational resources and possible improvements in the linear solvers.

In figure 10 we present measurements of the local expectation value $\langle n|S_i^z|n\rangle$ as a function of position in a $L = 26$ chain, for three values of disorder ($h = 1$ in the ETH phase, $h = 5$ in the MBL phase and $h = 3.8$, close to the MBL transition) and different eigenstates $|n\rangle$ in the middle of the spectrum. At low disorder, the local magnetizations are identical (with very small fluctuations) and small for all eigenstates: this sample is in the ETH phase. At large disorder $h = 5.0$, for a given eigenstate a majority of spins is polarized to their maximum value $|\langle n|S_i^z|n\rangle| \simeq 1/2$. The actual sign of polarization differs from one eigenstate to another, in perfect agreement with this disorder strength belonging to the MBL phase. We also see that the expectation value for some spins differ from this maximum polarization, and this for all eigenstates, meaning that hybridization has started to take place away from the infinite disorder limit $h \to \infty$ where all spins are polarized in all eigenstates. Qualitatively, the data at $h = 3.8$ are similar to those at the larger field $h = 5$. Quantitatively, a smaller number of sites and eigenstates have their absolute value of local magnetization close to $1/2$. With all precautions due to the use of a single sample and assuming that $h = 3.8$ is close enough to the true transition point $h_c$, this points towards the transition point not being thermal and not satisfying ETH.

Since we cannot perform computations in double precision for these large sizes, we consider the reliability of these single precision results by performing the rotation of nearby eigen-

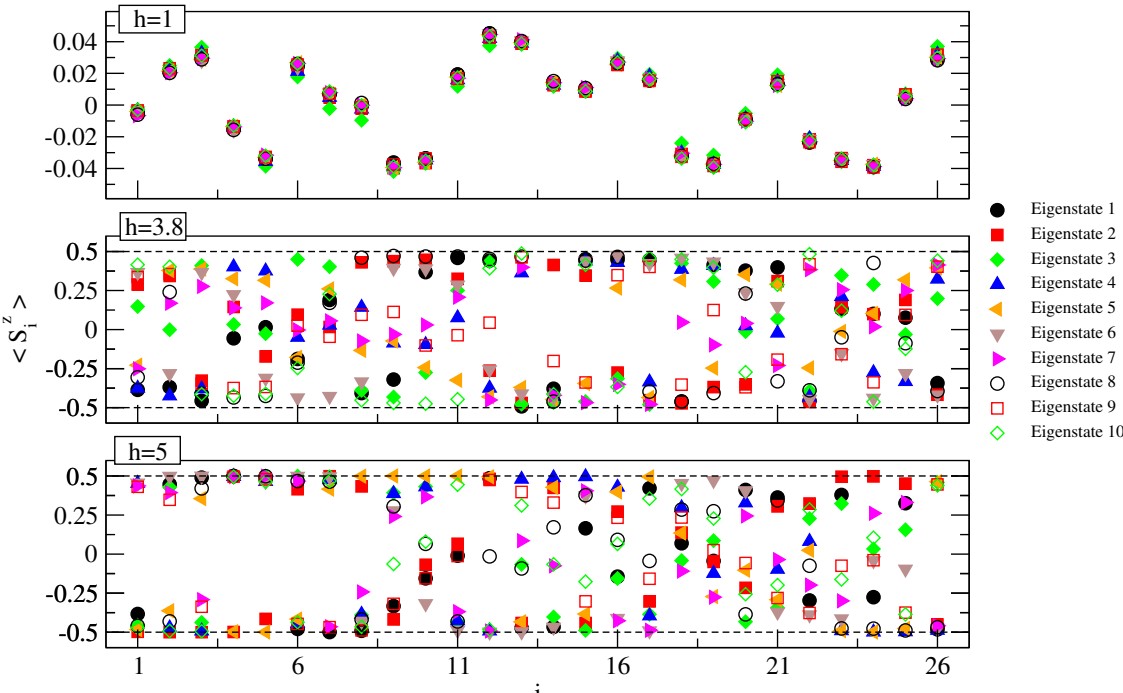

Figure 10: $\langle S_i^z \rangle$ as a function of site index $i$ for a $L = 26$ chain for 10 eigenstates at $\epsilon = 0.5$ with different values of disorder: $h = 1$ (top panel) in the ETH phase, $h = 5$ (bottom) in the MBL phase, and $h = 3.8$ (middle panel) close to the critical point.

pairs in order to minimize entanglement entropy, as discussed in Sec. 4.4. We found some instances where a very small rotation was necessary, but this has essentially no effect on the local magnetization values $\langle S_i^z \rangle$ presented above.

## 6   Discussion and conclusion

We discussed the shift-invert method for the MBL problem, and presented an analysis of its efficiency, pushing the method to its limits. As it stands now, it is the most efficient technique to obtain exact, unbiased, interior eigenpairs in all the possible regimes of this many-body problem. Data obtained on larger systems than previously available ($L = 24 - 26$) confirm the presence of a critical point $h_c \simeq 3.8$ (we defer a precise and complete finite-size scaling analysis of our new data to a future work).

The shift-invert technique is used in several fields of science, and we briefly discuss now related applications in condensed matter physics. Shift-invert is also the state of the art method to obtain eigenstates for the closely related problem of single-particle Anderson localization [28]. Here, for a linear system size $L$ in dimension $d$, the size of the matrix is $L^d$. Typical state of the art calculations in $d = 3$ reach $L \simeq 120$ to $150$ [29,30], and $L \simeq 1000$ for related $d = 2$ computations of quantum Hall transitions [31,32]. The record computation for the $d = 3$ Anderson problem is for a $L = 250$ system (matrix size $1.56 \cdot 10^7$, with in total $9.4 \cdot 10^7$ off-diagonal elements) in Ref. [33], in the localized phase or close to the transition point. This is slightly smaller than the largest system we studied here, $L = 26$ (with Hilbert space size $10^7$ and $1.4 \cdot 10^9$ off-diagonal elements). For the Anderson problem, the best performances [33] are obtained not by solving directly Eq. (4) but rather by using iterative solvers with incomplete LU factorization and advanced preconditioning techniques. A similar strategy as the one advocated here (direct parallel LU solver) was also used successfully in

electronic structure calculations in Ref. [49], with matrices (obtained from density-functional based tight-binding) of size up to 512000 with $\sim 4 \cdot 10^7$ off-diagonal elements, albeit asking for a much larger number of eigenpairs.

Besides scaling the computational resources and potential improvements in direct solvers, how can we possibly reach interior eigenpairs for even larger systems? Filtering methods, such as with Chebyshev polynomials [50] or rational functions [51, 52], offer an interesting alternative to the shift-invert procedure described here. Large scale results have been obtained using Chebyshev polynomials [50] on matrices larger than those presented here. However the exponentially small energy level spacing in the middle of the spectrum for MBL might render these methods inefficient. Another possibility could be to use incomplete LU factorization and preconditioning in order to solve iteratively the linear system Eq. (4), as for the Anderson problem [33]. We tested several iterative solvers with different types of preconditioning, as available through PETSc, but did not find superior efficiency compared to the direct solvers. It could nevertheless well be that a better pre-conditioning based on heuristics and physical knowledge on the eigenstates can improve the situation, in particular at large disorder where the local integral of motions picture [53–55] for eigenstates is available. Based on our experience with incomplete LU (ILU) preconditioning, we speculate that the significantly increased (extensive) number of nonzero matrix elements per row of the Hamiltonian in the MBL problem compared to the 3d Anderson case makes low level ILU preconditioning ineffective and a deeper level is required, at which point it becomes the full LU decomposition we use.

Finally, there are several other models besides Eq. (1) which display a MBL phase, and it is also possible that direct or iterative solvers might allow to reach even larger matrix sizes than for the random-field Heisenberg chain studied in this work. Also the best performing LU solver may depend on the model, and can be determined by simulations on smaller/intermediate sizes in the same spirit as we presented here. In our case, we found useful for both efficiency tests and production runs to distinguish small problems (which fit one one node) from intermediate to large problems requiring several nodes.

As for computational methods, we find that MBL is a hard and challenging application for direct solvers and interior eigenvalues methods, in particular due to the large fill-in observed in the factorization step. It would be interesting to find adapted re-ordering strategies that could reduce the fill-in, allowing to reach larger sizes. Finding a simple, unique, successful re-ordering strategy could turn out to be difficult as the structure of the graph generated by the Hamiltonian is quite complex, and is size-dependent (we tried several other ad-hoc re-ordering schemes, with at best very limited gains). The source code presented in the appendix allows to save the matrix in the Matrix Market exchange format[5] for future tests to be performed.

# Acknowledgements

We are very grateful to the developers of the various linear algebra libraries used in this work, and acknowledge in particular very fruitful and instructive exchanges with Patrick Amestoy, Alfredo Buttari and Pieter Ghysels.

**Funding information** This work benefited from the support of the project THERMOLOC ANR-16-CE30-0023-02 of the French National Research Agency (ANR) and by the French Programme Investissements d'Avenir under the program ANR-11-IDEX-0002-02, reference ANR-10-LABX-0037-NEXT. This project also received funding from the European Union's Horizon 2020 research and innovation programme under the Marie Skłodowska-Curie grant agreement No. 747914 (QMBDyn). We acknowledge PRACE for awarding access to HLRS's Hazel Hen

---

[5]https://math.nist.gov/MatrixMarket/

computer based in Stuttgart, Germany under grant number 2016153659, as well as the use of HPC resources from CALMIP (supercomputer eos, grants 2017-P0677 and 2018-P0677) and GENCI (supercomputers ada and occigen used for testing, grant x2018050225).

**Numerical libraries**   We use the following versions of the linear algebra libraries: PETSc (version 3.8.2), SLEPc (version 3.8.2), MUMPS (version 5.1.2), STRUMPACK (version 2.1.0), METIS (version 5.1.0), and the Intel C++ compiler version 17.1.0.

# A   Shift-invert example code

Here we briefly outline the basic structure of a C++ example code performing the shift-invert diagonalization of the Hamiltonian (1). The source code can be found online as a git repository under the URL https://bitbucket.org/dluitz/sinvert_mbl, licensed under the GPL v3.

## A.1   Setup

In order to get the code to run, the following steps have to be performed (in order). Note that we assume a standard Linux system here and indicate these steps only as an example. For your system, please refer to the documentations of PETSc and SLEPc.

1. Download PETSc. The easiest way to do so is by cloning the git repository and checking out the latest stable version (here we will use version v3.8.2).

```
$> mkdir petsc_dir/
$> cd petsc_dir/
$> git clone https://bitbucket.org/petsc/petsc.git .
$> git fetch --all --tags --prune
$> git checkout tags/v3.8.2
```

2. Next, PETSc has to be configured and compiled:

```
$> cd petsc_dir/
$> export PETSC_DIR=$PWD/
$> export PETSC_ARCH=linux-real-mumps
# configure petsc with mumps, scalapack and metis
$> ./configure --download-scalapack --download-metis
   --download-mumps
$> make all
```

3. Now, SLEPc can be downloaded and installed:

```
$> mkdir slepc_dir/
$> cd slepc_dir/
$> git clone https://bitbucket.org/slepc/slepc .
$> git fetch --all --tags --prune
$> git checkout tags/v3.8.2
$> export SLEPC_DIR=$PWD/
$> ./configure
$> make all
```

4. Now, the example code can be downloaded and compiled. Note that we use a setup file located in `conf/linux.cmake`, which should be modified if you use different li-

braries/settings on your system. It is then automatically read by `cmake` using the
`-DMACHINE=linux` flag.

```
$> mkdir sinvert/
$> cd sinvert/
$> git clone https://bitbucket.org/dluitz/sinvert_mbl .
$> mkdir build
$> cd build
$> cmake -DMACHINE=linux ../src
$> make
```

5. Finally, an options file with the name `slepc.options` has to be created and placed in
the same path as the executable; for example, one could use the following settings:

```
$> cat slepc.options
   -L 16
   -nup 8
   -Delta 1.3
   -W 3.
   -eps_type krylovschur
   -eps_nev 20
   -st_type sinvert
   -st_ksp_type preonly
   -st_pc_type lu
   -st_pc_factor_mat_solver_package mumps
```

6. Now you can run the program for example using 4 MPI processes using

```
$> mpirun -np 4 ./sinvert

   [Basis] generating basis tables for qbit chain of size
      16 with nup conservation.
   [Basis] the number of basis states is 12870
   [Hamiltonian] created with Delta = 1.
   [Hamiltonian]           J = 1.
   [Hamiltonian]           L = 16.
   [Hamiltonian]           fields = 0.557068 2.06559
      2.14767 2.08351 0.741382 -0.69371 -1.21479 -2.65972
      -1.36406 -0.134009 1.87301 -0.120137 -0.643291
      2.01647 -0.975623 0.889031
    Storing matrix...  done.
    Assembly... done.
         E1 = 11.7576
         E0 = -11.1217
   ----------------------------------
   Central eigenpairs:
   E(0) =0.31798568877554  <psi|Sz[3]|psi> =
      -0.397612088244758
   E(1) =0.318041919831143  <psi|Sz[3]|psi> =
      0.0809825878586985
   E(2) =0.3175376516457  <psi|Sz[3]|psi> =
      -0.00151934755149076
   E(3) =0.317295960240257  <psi|Sz[3]|psi> =
      0.00825505532713932
   E(4) =0.319131832142374  <psi|Sz[3]|psi> =
      0.0243165969903161
```

```
            E(5)  =0.316492528819752  <psi|Sz[3]|psi> =
                0.0101557163041098
            E(6)  =0.320017118229579  <psi|Sz[3]|psi> =
                0.097939842077515
            E(7)  =0.320744629655732  <psi|Sz[3]|psi> =
                -0.196587758708495
            E(8)  =0.315040905320559  <psi|Sz[3]|psi> =
                -0.152409388811942
            E(9)  =0.320855685192556  <psi|Sz[3]|psi> =
                0.271853007185108
            E(10) =0.314149256014437  <psi|Sz[3]|psi> =
                0.180480868425142
            E(11) =0.314102518287276  <psi|Sz[3]|psi> =
                0.009500296943451
            E(12) =0.313583616416597  <psi|Sz[3]|psi> =
                -0.05383114372309
            E(13) =0.322569170802922  <psi|Sz[3]|psi> =
                -0.103092269894268
            E(14) =0.323083433394346  <psi|Sz[3]|psi> =
                0.0374621752963861
            E(15) =0.312379224447301  <psi|Sz[3]|psi> =
                0.0913555438542086
            E(16) =0.323597512559935  <psi|Sz[3]|psi> =
                0.0279025347789937
            E(17) =0.312109535398481  <psi|Sz[3]|psi> =
                0.0660436092986919
            E(18) =0.32404678833222   <psi|Sz[3]|psi> =
                -0.063519907123315
            E(19) =0.311476024662365  <psi|Sz[3]|psi> =
                0.0242486268077492
```

## A.2 Basis generation

Since the accessible system sizes for shift-invert diagonalization are relatively modest compared to ground state methods, it is sufficient to use an encoding of the computational basis of the Hilbert space which is based on full enumeration. We use the basis where the local $S_i^z$ operators on site $i$ are diagonal and therefore the basis states are given by product states labeled by the eigenvalues $\sigma_i = \uparrow, \downarrow$ of the $S_i^z$ operators. Each state can efficiently be encoded as a bit string of $L$ bits, where $L$ is the length of the spin chain making the identification $0 \rightarrow \downarrow$ and $1 \rightarrow \uparrow$. The Basis class is implemented in the source file Basis.h. We provide two implementations of the basis, one is the full basis of all possible $S^z$ product states for size $L$, constructed by Basis(size_t L), the second one is the full basis of a sector of the Hilbert space with a fixed number of up spins $n_\uparrow$, which is useful for problems with conservation of magnetization. It is constructed by Basis(size_t L, size_t nup). Here is the basic interface of the Basis class (see source code for the full interface):

```cpp
typedef std::bitset<NBITS> State; // State coded as bitstring
class Basis // bit coded basis for qbit chains of length L
{
   private:
      bool conserve_nup; // flag to switch on/off the conservation of the
         number of up spins
      size_t L; // length of the spin chain
      size_t nup; // number of up spins (only used if conserved)
      size_t size;
```

```
        std::vector<State> basis_states; // contains the basis states
            ordered by their index
        std::vector<unsigned long> state_indices; // only works for up to
            64 bits! Contains indices of basis states ordered by their
            binary representation for inverse lookup
    public:
        Basis(size_t L_, size_t nup_); // constructor for Basis with fixed
            nup (sets conserv_nup to true)
        Basis(size_t L_); // constructor for full basis

        size_t get_index(State state);
        State get_state(size_t index);
};
```

An instance of the basis class has to be generated before constructing the Hamiltonian object, and handed to its constructor as a pointer. It is used to look up the index of basis states which are generated after application of terms of the Hamiltonian.

### A.3  Hamiltonian generation

The generation of the sparse Hamiltonian matrix H has to be performed in parallel for large system sizes, in particular because the matrix is stored in a distributed fashion using the sparse matrix module of PETSc. We use two steps: The first step `calculate_nnz` is a dry-run, counting only the number of nonzero matrix elements generated later by `calculate_sparse_rows` for an optimal memory allocation.

```
class XXZHamiltonian
{
    private:
        Basis * baseptr; // pointer to basis
        size_t L; // length of the chain (inferred from Basis)
        double Delta; // Sz_i Sz_{i+1} prefactor
        double J; // S+_i S-_{i+1} prefactor
        std::vector<double> fields; // random fields coupled to Sz_i
    public:
        XXZHamiltonian(Basis * baseptr__, double Delta__, double J__,
            std::vector<double> fields__); // constructor
        void calculate_sparse_rows(size_t begin_row_, size_t
            end_row_,std::vector<int> & row_idxs_,
        std::vector<int> & col_idxs_, std::vector<double> & entries_); //
            calculate indices and entries of nonzero matrix elements in rows
            begin_row_ to end_row_
        int calculate_nnz(size_t begin_row_, size_t end_row_,
            std::vector<int> & d_nnz_,
        std::vector<int> & o_nnz_); // count number of block diagonal and
            offdiagonal nonzero matrix elements corresponding to MPI
            partition of the matrix (important for efficient PETSC matrix
            assembly)
};
```

### A.4  Shift-invert diagonalization with SLEPc

Here we show the important parts of the actual diagonalization code. The options are loaded from the `slepc.options` file, the basis and the Hamiltonian matrix are generated, and finally

the shift-invert diagonalization is performed by calling SLEPc's EPSSolve.

```cpp
SlepcInitialize(&argc, &argv, "slepc.options", help);
Basis basis(L, nup); // generate basis using nup conservation law
XXZHamiltonian hamiltonian(&basis, Delta, J, fields);
nnz = hamiltonian.calculate_nnz(Istart, Iend, d_nnz, o_nnz); //
    Preparation run, analyzing number of nonzero elements
hamiltonian.calculate_sparse_rows(Istart, Iend, rows, cols, entries);
Mat H;
MatCreateAIJ(PETSC_COMM_WORLD, Iend-Istart, PETSC_DECIDE,
    basis.get_size(), basis.get_size(), 0, d_nnz.data(), 0, o_nnz.data(),
    &H);
MatSetUp(H);
for(size_t i=0; i<entries.size(); i++)
{
    MatSetValue(H, rows[i], cols[i], entries[i], INSERT_VALUES);
}

MatAssemblyBegin(H, MAT_FINAL_ASSEMBLY);
MatAssemblyEnd(H, MAT_FINAL_ASSEMBLY);

// Shift-invert part
EPS eps_si;
EPSCreate(PETSC_COMM_WORLD, &eps_si);
EPSSetOperators(eps_si, H, NULL);
EPSSetProblemType(eps_si, EPS_HEP);
EPSSetFromOptions(eps_si);

EPSSetWhichEigenpairs(eps_si, EPS_TARGET_REAL);
EPSSetTarget(eps_si, (E0+E1)/2. );

EPSSolve(eps_si);
EPSGetConverged(eps_si, &nconv);

Vec evecreal, tmp;
MatCreateVecs(H, &evecreal, NULL);
MatCreateVecs(H, &tmp, NULL);

Operator op(&basis);

for(int i=0; i<nconv; i++)
{
    EPSGetEigenpair(eps_si,i, &kr, &ki, evecreal, tmp);
    VecCopy(evecreal, tmp);
    size_t operator_site=3;
    op.apply_Siz(operator_site, &tmp);
    double Sz;
    VecDot(evecreal, tmp, &Sz);

    if(0==myrank) std::cout << "E("<<i<<") = " << kr << "
        \t<psi|Sz["<<operator_site<<"]|psi> = " << Sz << std::endl;
}
```

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
