# Peer review of "Shift-invert diagonalization of large many-body localizing spin chains"

_SciPost Physics, doi:SciPost Phys. 5, 045 (2018)_

## Round 2 · Referee Report · Anonymous · 2018-8-5

Strengths

1- Very accessible introduction to state of the art exact diagonalization numerics
for the field-disordered Heisenberg chain.

2- This paper explains in detail how the impressive simulations contained in previous publications by some of the authors have been achieved, and enables a broader range of scientist to use these methods for similar problems, disordered or not.

3- Link to code is contained in the manuscript

Weaknesses

no particular weaknesses

Report

This manuscript explains in detail how the impressive simulations contained in previous publications by some of the authors have been achieved, and also pushes them to the currently possible limit of L=24 to L=26 spins depending on the precision. The reader is able to understand what libraries have been used, what has been tested and what the authors recommend to tackle these large-scale diagonalizations, which constitute the current state of the art in unbiased numerics for the MBL problem.

Requested changes

1- On page 4 a sector Sz=1 is mentioned. I believe this should be the Sz=1/2 sector for odd length chains

2- on page 8 there is a type "bot" -> "not"

3- on page 13, Fig. 5 suffers from an impaired font on my screen and in print.

---

## Round 2 · Referee Report · Anonymous · 2018-8-15

Strengths

1- Technical details on the shift-invert method.

2- Benchmarks of the shift-invert method applied to the many-body localization problem.

3- Some new data for bigger systems than published previously.

Weaknesses

1- No new physics.

2- Not always as pedagogical as intended.

3- A number of minor oversights in the presentation.

Report

The many-body localization problem, i.e., the phenomenon of a transition to localization in highly excited states of a quantum many-body system, has attracted a significant amount of interest over the past decade. The
present authors have made important contributions to the field using in particular "exact diagonalization". An important challenge in the many-body problem is the exponential growth of the Hilbert space dimension with the number of sites $L$ under consideration. For a spin-1/2 model with more than $L=18$ sites, the authors advocate the use of the "shift-invert" method in order to compute highly excited states for up to $L=26$ spins 1/2.

In the present manuscript, the authors give technical details on the application of the shift-invert method to the many-body localization problem and provide benchmarks. They also present some results on bigger systems than studied previously, specifically $L=24$ (section 5.1) and even
some partial results for $L=25$ and $26$ (section 5.2). However, the authors postpone a detailed analysis of these new results to future work (see, e.g., first paragraph of the Discussion, chapter 6). Consequently, there is no real new physics in the present paper. Rather, the present
manuscript is intended to be a "pedagogical review" (first line of abstract).

While such a technical background is interesting and deserves publication in principle, I feel that the manuscript requires a bit more work. In particular, the authors do not always live up to their pedagogical ambitions. Some concrete comments and suggestions for improving the manuscript are contained in the list below under "Requested changes" (order corresponding to order of appearance in the manuscript, by no means to importance).

In addition, I have one suggestion that probably goes beyond the scope of the present work. If single-precision arithmetics performs so well (section 4.4), one might ask if this could be combined with double-precision arithmetics in a post-processing step. In particular, since eigenvectors
are computed, one could double-check if the off-diagonal matrix elements of ${\sf H}$ between states that cannot be separated in single precision are non-zero in double precision, and if necessary perform a rotation of the basis vectors to diagonalize ${\sf H}$ in this quasi-degenerate subspace. Exploration of such alternatives to the two heuristic approaches proposed at the end of page 14 might be an interesting topic for future work.

Requested changes

1- Eq. (1): Here the Hamiltonian "$H$" is introduced with a different font than later in the text. If this was done intentionally, a comment would be appropriate, otherwise this typesetting error should be corrected.

2- Two lines below Eq. (1): the authors immediately specialize the anisotropy parameter $\Delta$ to the value $\Delta=1$. Without the local fields, i.e., for $h_j=0$, this would be a very special case with enhanced SU(2) symmetry. Hence, I think that a comment would be useful whether the authors expect the value $\Delta=1$ to correspond to a special or rather a
generic situation for the many-body localization problem.

3- Likewise, three lines below Eq. (1), the authors specialize the local fields to be drawn from a box distribution. It would again be interesting to know if the authors expect this distribution to be "universal", i.e., to exhibit behavior representative of generic field distributions.

4- The comment on the Jordan-Wigner transformation at the top of page 4 is meaningful only to a specialist reader. I am aware that a more detailed explanation goes beyond the scope of the present manuscript, but at least a reference would probably be helpful for the non-specialist reader. Note
also that it may be useful to state that the "spinless fermions" are interacting.

5- Section 3.1 has only one reference and this is Anderson's original work on the "Anderson" localization. I think that the authors should add some references for diagonalization algorithms. Actually, it is clear from the Discussion in chapter 6 that the authors are aware of relevant references. However, they should not postpone references to chapter 6, but at least cite Refs. [42-47] also in section 3.1.

6- A typesetting detail: why does footnote 3 not appear before page 10 ?

7- End of page 8: since the authors present CPU times, they need to specify the compute system exactly, most notably quote the CPU model and network interconnect.

8- In the line below Fig. 2 there is a reference to "Table 4.1", but no such table exists (same at the end of page 11).

9- Figure 3: Firstly, panel labels "(a)" and "(b)" should be added. Secondly, I did not managed to identify "the thick line" mentioned in the caption.

10- Figure 5: Black font on a dark blue background is difficult to read. Maybe the authors should use a white font for "16".

11- Figure 7: Clarity of the left panel would probably be improved if the data for $h=1$ and $100$ would be presented in separate panels.

12- Last paragraph of section 5.1: without consulting Ref. [41], it is not clear with respect to which variable the "slope" of the entropy $S$ is computed. In a presentation that pretends to be a review, I think that the authors should add the necessary explanation rather than sending the reader to the literature.

13- Figure 10: Firstly, this figure seems not to be cited at all in the text. Secondly, the caption refers to "3 eigenstates" whereas according to the legend, the actual number seems to be 10.

14- The last sentence of the Discussion (chapter 6) did not make sense to me. Maybe this is because the term "Matrix Market" would need explanation.

15- In the "Numerical libraries", the authors specify version 3.8.2 of PETSc and SLEPc. However, a little further down in Appendix A.1, they use versions 3.7.7 and 3.7.4, respectively. This should be harmonized.

16- References need to be proof-read carefully. Proper names such as "Hamilton(ians)", "Schur", "Anderson", and "Chebyshev" should be in upper case (most of the time they are not), there is garbage in the initials of the second author of Ref. [19], and "XXZ" in the title of Ref. [19] should probably also be upper case.

---

## Round 2 · Referee Report · Anonymous · 2018-8-23

Strengths

1- nice pedagogical review for the shift-inevert method for the exact diagonalization

2- detailed examination of the efficiency of the shift-invert method

Weaknesses

no weakness

Report

In this paper, the authors explain the efficient shift-invert technique to calculate the excited states for the random Heisenberg model.
They examine the efficiency of the shift-invert computations by using the several different solvers for linear equations. They also apply the method for large MBL systems and obtain insight in the MBL problems.

The example code and the detailed examination of the
shift-invert method are useful for the researchers in
the field of the condensed matter physics and the computational science.
Thus, I recommend the publication of the paper
in the SciPost Physics.

Requested changes

1- several typos in the manuscript. e.g.,
page. 3 denotes the strenght -> denotes the strength

---

## Round 3 · Author Response

Dear editor,

thank you for arranging the review of our manuscript. We would like to thank the referees for their comprehensive reports which have helped us to further improve it, and we are happy that our work is perceived as interesting and useful.

To address one of the remarks raised in report 2, we specifically improved the introduction to make it more pedagogical, performing the changes requested by the referee. Please note that this paper is also accompanied by a well-commented and documented code, which we trust will make it especially clear and helpful for researchers aiming to learn this numerical method.

In the updated version of our manuscript we have carefully addressed all the suggested changes; below we reply in detail to the various points raised in the three referee reports highlighting the changes made. We also performed very minor reformulations in the body of the text.

We hope that with these improvements, you will judge our manuscript suitable for publication in SciPost Physics.

Yours sincerely, The authors

Reply to Referees

Reply to Referee 1

We thank the referee for his/her positive appreciation of our work.

Ref. 1: On page 4 a sector Sz=1 is mentioned. I believe this should be the Sz=1/2 sector for odd length chains Our reply: The referee is correct and we have changed $S_z=1$ to $1/2$.

Ref. 1: on page 13, Fig. 5 suffers from an impaired font on my screen and in print Our reply: We have improved Figure 5 with respect to the fonts and colors used.

Ref. 1: on page 8 there is a type "bot" $\to$ "not" Our reply: We thank the referee and have corrected this typo.

Reply to Referee 2

We thank the referee for his/her overall positive appreciation of our work, and for the suggestions for improving our manuscript. We discuss them in detail below. Regarding the pedagogical nature of this work and besides the improvements discussed below, we would like to point out that our manuscript is accompanied by a well-commented and documented code, which we trust will make it especially clear and helpful for researchers aiming to learn this numerical method.

Regarding the specific changes requested:

  • Ref. 2: Eq. (1): Here the Hamiltonian ``H" is introduced with a different font than later in the text. If this was done intentionally, a comment would be appropriate, otherwise this typesetting error should be corrected.

    Our reply: We thank the referee, this was indeed a typesetting error. We now use the same symbol everywhere for the Hamiltonian.

  • Ref. 2: Two lines below Eq. (1): the authors immediately specialize the anisotropy parameter $\Delta$ to the value $\Delta=1$. Without the local fields, i.e., for hj=0, this would be a very special case with enhanced SU(2) symmetry. Hence, I think that a comment would be useful whether the authors expect the value $\Delta=1$ to correspond to a special or rather a generic situation for the many-body localization problem.

    Our reply: We have clarified here that $\Delta>0$ is a generic representative of a system with an MBL transition. Also please note that any non-zero value for the random field breaks the SU(2) symmetry, as well as the integrability of the clean XXZ chain, as can be seen e.g. in the statistical properties of energy eigenvalues.

  • Ref. 2: Likewise, three lines below Eq. (1), the authors specialize the local fields to be drawn from a box distribution. It would again be interesting to know if the authors expect this distribution to be "universal", i.e., to exhibit behavior representative of generic field distributions.

    Our reply: The specific choice of disorder distribution is not crucial for the purpose of this paper, which mainly focuses on numerical methods. We added a comment to this effect. We note however that other distributions (e.g. quasiperiodic or binary) may lead to different physical properties.

  • Ref. 2: The comment on the Jordan-Wigner transformation at the top of page 4 is meaningful only to a specialist reader. I am aware that a more detailed explanation goes beyond the scope of the present manuscript, but at least a reference would probably be helpful for the non-specialist reader. Note also that it may be useful to state that the "spinless fermions" are interacting.

    Our reply: We removed the mention to the Jordan-Wigner transformation since it is not central here and would be meaningful only to specialists.

  • Ref. 2: Section 3.1 has only one reference and this is Anderson's original work on the "Anderson" localization. I think that the authors should add some references for diagonalization algorithms. Actually, it is clear from the Discussion in chapter 6 that the authors are aware of relevant references. However, they should not postpone references to chapter 6, but at least cite Refs. [42-47] also in section 3.1.

    Our reply: We have reorganized the citations, which now appear in section 3 rather than only in section 6.

  • Ref. 2: A typesetting detail: why does footnote 3 not appear before page 10?

    Our reply: We have fixed this typesetting anomaly.

  • Ref. 2: End of page 8: since the authors present CPU times, they need to specify the compute system exactly, most notably quote the CPU model and network interconnect.

    Our reply: We agree with the referee and we now mention the hardware of both computing systems used.

  • Ref. 2: In the line below Fig. 2 there is a reference to "Table 4.1", but no such table exists (same at the end of page 11).

    Our reply: We have corrected these references to correctly point to table 2.

  • Ref. 2: Figure 3: Firstly, panel labels "(a)" and "(b)" should be added. Secondly, I did not managed to identify "the thick line" mentioned in the caption.

    Our reply: We added the (a) and (b) labels and further marked the `thick line' in black.

  • Ref. 2: Figure 5: Black font on a dark blue background is difficult to read. Maybe the authors should use a white font for "16".

    Our reply: We have improved the fonts in figure 5 (see also Reply to Referee 1).

  • Ref. 2: Figure 7: Clarity of the left panel would probably be improved if the data for h=1 and 100 would be presented in separate panels.

    Our reply: We understand the referee's motivation for this comment. However, it is our opinion that separating the left panel of figure 7 in two panels (like in the right panel) would decrease the readability of the plot in this case. Also we believe it helps the reader realizing that the ETH (at $h=1$) and MBL ($h=100$) phases have clear distinct behaviors for physical observables.

  • Ref. 2: Last paragraph of section 5.1: without consulting Ref. [41], it is not clear with respect to which variable the "slope" of the entropy S is computed. In a presentation that pretends to be a review, I think that the authors should add the necessary explanation rather than sending the reader to the literature.

    Our reply: We have added a clear definition of the entanglement entropy slope with respect to subsystem size, to make this section self explanatory without consulting the literature.

  • Ref. 2: Figure 10: Firstly, this figure seems not to be cited at all in the text. Secondly, the caption refers to "3 eigenstates" whereas according to the legend, the actual number seems to be 10.

    Our reply: The referee is right and we have corrected the caption. Additionally, we cited this figure in the text (section 5.2).

  • Ref. 2: The last sentence of the Discussion (chapter 6) did not make sense to me. Maybe this is because the term "Matrix Market" would need explanation.

    Our reply: The Matrix Market exchange format is a standard for files storing matrices (see https://en.wikipedia.org/wiki/Matrix_Market_exchange_formats ). We modified the corresponding sentence and added a footnote in order to make it clear that Matrix Market refers to this exchange file format.

  • Ref. 2: In the "Numerical libraries", the authors specify version 3.8.2 of PETSc and SLEPc. However, a little further down in Appendix A.1, they use versions 3.7.7 and 3.7.4, respectively. This should be harmonized.

    Our reply: After proper testing, we now mention versions 3.8.2 as the reference PETSc and SLEPc versions in the appendix.

  • Ref. 2: References need to be proof-read carefully. Proper names such as "Hamilton(ians)", "Schur", "Anderson", and "Chebyshev" should be in upper case (most of the time they are not), there is garbage in the initials of the second author of Ref. [19], and "XXZ" in the title of Ref. [19] should probably also be upper case.

    Our reply: We corrected the wrong capitalizations in the bibliography style.

  • Ref. 2: In addition, I have one suggestion that probably goes beyond the scope of the present work. If single-precision arithmetics performs so well (section 4.4), one might ask if this could be combined with double-precision arithmetics in a post-processing step. In particular, since eigenvectors are computed, one could double-check if the off-diagonal matrix elements of H between states that cannot be separated in single precision are non-zero in double precision, and if necessary perform a rotation of the basis vectors to diagonalize H in this quasi-degenerate subspace. Exploration of such alternatives to the two heuristic approaches proposed at the end of page 14 might be an interesting topic for future work.

    Our reply: We thank the referee for this interesting suggestion. Following it, we have performed numerical experiments which appear to indicate that the knowledge of single-precision eigenvectors spanning the quasi-degenerate subspace is not sufficient to accurately reconstruct the double-precision non-zero off-diagonal elements, making it impossible to find the rotation yielding the correct eigenvectors. For that reason, to determine the correct eigenvectors from single-precision data, we still resort to the minimum of entropy method exposed in the article.

Reply to Referee 3

We thank the referee for his/her positive appreciation of our work. We reviewed the draft carefully for typos and corrected them.

---

## Round 3 · List of Changes

Besides minor typos, all the changes correspond to precisions asked by the referees, in particular Ref. 2.
See above for a precise list of the changes made.

---

## Editorial Decision

published